# 'Make Your House like a Temple': Gender, Space and Domestic Devotion in Medieval Florence

Catherine Lawless

Trinity College Dublin, The University of Dublin, Dublin 2, Ireland; lawlessc@tcd.ie

**Abstract:** This article will discuss domestic devotions by framing them in terms of devotions carried out in the home, defined by its opposition to ecclesiastical, consecrated space. It will examine how women, considered the laity par excellence through their inability to ever attain sacerdotal authority, were advised spiritually by mendicant friars on how to lead a Christian life according to their status as wives, widows or virgins. It will look at the devotional literature that was widespread in mercantile homes and the devotional images designed to move the soul. This discussion will attempt to show the tensions between ecclesiastical and domestic spaces; between the clergy and the laity, and between the corporeal and spiritual worlds of late medieval devotion. It will argue that, despite clerical unease with the female and domestic space, the importance accorded to female piety by the mendicant orders at the close of the Middle Ages was such that women were entrusted with key educational roles in the family, even leading to the astonishing affirmation of them as 'preachers' within the borders of their households.

**Keywords:** mendicants; devotional texts; devotional art; female patronage; Florentine trecento

## 1. Introduction

In a well-known treatise, *Regola del governo di cura familiare*, written in 1403, the Dominican friar Giovanni Dominici advised Bartolomea degli Obizzi, widow of the exiled Antonio di Niccolò degli Alberti, on how to practice devotion within the home and raise her four children (Dominici 1860, p. 132; Baxendale 1991, p. 731). After having advised her to have paintings in the house to instruct the children, as 'paintings of Angels and Saints are allowed and ordered, for the lower mental levels'[1], he went on to say that if she did not want to have images, or could not have, 'such paintings to make almost a temple of the home' she should have the children frequently brought to church when offices were not said, so they should become used to holy places.[2] The polarity between the ecclesiastical and liturgical space of the church and the domestic one of the home was ideally to be overcome by the transformation of the home into a church, and the (male) children were to be taught the principles of sacramental religion by being urged to dress as priests, to create little altars, to decorate them, clean them and adorn them with images (Dominici 1860, p. 146).[3] There is no doubt about the superiority of the ecclesiastical space

---

[1] 'Però che debbi sapere sono permesse e ordinate le dipinture degli Angeli e Santi, per utilità mentale de' più bassi'.

[2] 'E se non vuogli, o non puoi, di tante dipinture fare quasi tempio in casa, avendo balia fa' sieno menati spesso in chiesa a tempo non vi sia tumulto, nè vi si dica ufficio; acciò nè lor mente sia rapita dalla tumultuata gente, nè lor cianciare dia impaccio al divino ufficio.'

[3] 'Ma farai uno altaruzzo o due in casa, sotto titolo del Salvatore, del quale è la festa ogni domenica: abbivi tre o quattro dossaluzzi variati, ed egli, o più, ne sieno sacrestani; mostrando loro come ogni festa debbano variatamente adornare quella cappelluzza. Alcuna volta saranno occupati in fare grillande di fiori o d'erbe, e incoronare Iesu, adornare la Vergine Maria dipinta, fare candeluzze, accendere e spegnere, incensare, tenere pulito, spazzare, parare gli altari, comporre de' candelieri di fuscelli di cera, di terra; sievi la campanuzzi, corrino a sonare tutte l'ore co me sentono nelle chiese, possansi parare con le camice come accoliti, cantinvi come sanno, parinsi a dir messa, e sieno menato alcuni volta alla ch iesa e loro mostrato quel

and only the boys can become its administrators: priests. The passage neatly illustrates the difference between the familial space, sometimes, and arguably, governed by the woman, and the ecclesiastical, governed by the male clergy.

In this contribution, I examine the domestic devotions of women as exemplifying those of a group that was lay *in extremis*; that is, unable ever to enter sacerdotal functions and spaces. Using the sermons and treatises of the Dominicans Giordano da Pisa (d.1310), Domenico Cavalca (d.1342), Giovanni Dominici (c.1355–1419), Antoninus Pierozzi (1389–1459) and the Franciscan Bernardino da Siena (1380–1444), I will examine the messages transmitted through these texts to women. I will then look at popular devotional texts that were demonstrably owned by women and devotional art originating, in all probability, due to scale, subject matter and format, from domestic spaces. The appearance of a female donor figure or a couple will further place these images within female and domestic concerns. The voices of women themselves remain isolated and sporadic, but I shall use that of Margherita Bandini Datini (c.1360–1423) wife of the famous merchant of Prato, Francesco di Marco Datini. Her voice is not to be discounted as an important witness of fourteenth-century life, as significant for relaying what her daily concerns were when negotiating the demands of her household and mediating between domestic life and religious requirements.

Medieval Christianity bore within it a number of tensions: those between the religious and secular, sacred and profane, male and female, virginal or celibate and married. These opposite poles were also nodes of power relations in which, in all cases, the dominant power (religious, sacred, male, virginal) was the former, the lesser was the latter (secular, profane, female, married). The highest form of life was the religious one, in which, theoretically, celibate male priests held the sole authority over the sacrament of the Eucharist, necessary, after 1215, for every Christian to consume at least once a year, in a suitable state of penitence after having been confessed (Von Moos 1996, p. 120; Rusconi 1981, pp. 68–69). Underlying my discussion is the contrast between masculine, consecrated space and its ministers, and domestic space, conceived of by these friars as feminine, or at least as a space in which women could exert significant influence. Ecclesiastical space was sharply distinct from domestic space.

Women, operating largely within a vernacular milieu, were usually responsible for the religious instruction of their children, and were seen as significant players in the late medieval piety, known as affective and penitential. In fourteenth- and fifteenth-century Florence, as elsewhere, women's spirituality and devout piety had long been recognized as powerful, but in need of regulation and guidance (Benvenuti Papi 1990, pp. 119, 205, 218, 242, 247). Of Cistercian inspiration, the penitential movement emphasized affectivity in devotional practice, and empathy in considering the Passion of Christ and the lives of his mother and the saints. Promoted effectively by the mendicant orders in the thirteenth and fourteenth centuries, the movement was particularly popular with women, already culturally linked with affectivity and the body (Bornstein 1996, p. 7; Vauchez 1993, p. 408; Bynum 1991). The literature on this feminized devotion is vast (Casagrande 1988, pp. 251–89; Vauchez 1990, pp. 206–20; Bynum 1991, pp. 181–238; Grundmann 1995; Bornstein 1996, pp. 1–27). Linked with mendicant promotion of the suffering humanity of Christ was the growth of vernaculars in the distribution and reception of religious texts (which naturally favoured women, among other groups of the 'unlettered') (Cornish 2000, pp. 166–80) and the spread of Eucharistic cults in the wake of the decrees of the Fourth Lateran Council of 1215, emphasizing transubstantiation, the requirement of the faithful to receive Communion at least once a year and to confess before communicating. As women's nature was seen as more pliable and receptive, female experiences as visionaries and mystics were, if accepted as orthodox in origin, welcomed by the clergy in the struggle against heresy and in the popularization of the Eucharist (Bynum 1991, pp. 183–87; Frugoni 1996, pp. 130–84). However, the widespread belief that women had weaker intellects and more fragile physical and mental natures assumed that they were

che fanno e' veri sacerdoti acciò imparino a contraffargli; e così variatamente quanto di può sieno occupati con amore circa il divino santuario, lasciandogli guastare le frascoline loro faranno, acciò abbin bisogno di rifarle.'

as susceptible to diabolical influence, thus their piety had to be managed and supervised by the clergy (Bornstein 1998, pp. 82–83).

## 2. The Spaces of Devotion

The difficulties in looking at domestic devotions begin with trying to define the term 'domestic' and what it may have meant in medieval and Renaissance Florence. Recent scholarly interest in the material culture of religious devotion in the home has enriched the historiography of medieval and renaissance social life (Lydecker 1987; Webb 1990; Musacchio 1997, 1999, 2008; Ajmar-Wollheim et al. 2006; Anderson 2007; Campbell et al. 2013; Brundin et al. 2018; Corry et al. 2017, 2019). The sacred place of the altar at which Mass is celebrated is differentiated in space from the domestic space, understood here as the dwelling place and culture of the family (often consisting of patrilineal groups rather than a single nuclear one) group and its household (which could include a number of servants), and mindful of the cautions indicated by Corry, Faini and Meneghin in articulating the fluidity of the borders of the domestic (Corry et al. 2019, p. 2). As observed by Marta Ajmar-Wollheim, Flora Dennis and Ann Matchette, the idea of the early modern house as a site of privacy and seclusion has dominated historical debate. The Victorian model of private and domestic being the opposite of public and non-domestic does not suit medieval and Renaissance Florence, where houses were often the sites of business, display and diplomacy as well as being homes (Ajmar-Wollheim et al. 2006, p. 623). Morse has written of the complexity of the Renaissance casa, warning that it could not be considered a secular space, but rather, a complex environment, situated at the intersection of public and private, individual and communal, spiritual and worldly (Morse 2007, p. 184). Although written about the Venetian casa, the same could be said of the Florentine palazzo in its mix of functions, with banks and shops often forming part of the house and with business being conducted within the home.

Although borders between domestic and non-domestic are fluid, and terms such as 'public', 'private', and 'personal' must be used with care, there were clear distinctions between ecclesiastical space and other spaces, including the home or indeed liturgical spaces (chapels) within the domestic space. The ceremony of consecration circumscribed a sacred place and made it holier than the surrounding area. Careful control was exercised over when and where the sacrifice of the Mass was celebrated and the conditions necessary for that process. Mass was celebrated in a distinct space, differentiated from the domestic, even when permission was obtained for the privilege of a private chapel (Böninger 2000, pp. 335–37). Jérôme Baschet points out that there is no great boundary between secular and sacred, but rather degrees of sacredness, radiating from poles such as the consecrated altar with the presence of the Eucharist (Baschet 2008, pp. 70–71). In his *I frutti della lingua*, a translation of St Gregory's dialogues and the letter of St Jerome to his female disciple Eustochium, the Dominican preacher Domenico Cavalca (d.1342) made it clear that sin committed in the church was worse than if it were done anywhere else, stating that greater reverence is due to the house of God in which the sacraments are administered, especially that of the 'precious body and blood' of Christ (Cavalca 1837, p. 74). Cavalca also discusses the laws of purification which command that a woman, after childbirth, through which 'certain corporeal dirt' is contracted, should not enter the temple until after forty days, when the dirt has ceased. (Cavalca 1837, p. 77). Fra Bernardino da Siena (1380–1444) stated a similar view when he argued that, although God is everywhere, he is more present in the church than elsewhere. Bernardino railed against those who misused this sacred space, who traded or chattered and he particularly noted those women who went there to see or be seen (Bernardino da Siena 1935, pp. 85–87), words echoed by Antoninus (Pierozzi 1858, p. 141) and the Dominican Fra Girolamo Savonarola (1450–1498) (Savonarola 1845, pp. 72–73).

## 3. Women's Space

Sacerdotal authority is confined to men and ecclesiastical authority, or 'the body of specialists of religion' (Bourdieu 1971, p. 304). Within the categories of the laity, in medieval Florence, men had greater access to spaces outside the home than women, or at least the elite women who were

the subject of behavioural manuals and who received tailored spiritual advice in the form of letters or treatises from mendicant friars. The gendered nature of the 'production of space' (Lefebvre 1991) has been explored for its implications in the medieval hierarchies allocated to burial and building by Roberta Gilchrist (Gilchrist 1999) and specifically in the Florentine context by Robert C. Davis (Davis 1998, pp. 19–38), Adrian Randolph (Randolph 1997, pp. 17–41) and Ena Giurescu Heller (Heller 2005, pp. 161–83). The role model for women was the Virgin Mary and, according to Paolo da Certaldo (fl. 1347–1370), a secular writer, women should, like her, stay indoors in a hidden and honest place (Paolo da Certaldo 1945, p. 59). The domestic space, as such, was not always 'hidden', with a workshop or shop often occupying the same building as the home (Cavallo 2006, p. 66), but women were rarely, if ever, occupants of those 'public' spaces. Women were forbidden to enter most civic buildings in Florence. They were not allowed, for instance, to enter the Palazzo del Podestà, for instance, having to instruct their notaries at its door (Laufenberg 2009, pp. 199–200).

The place for secular women was the home. As Adrian Randolph points out, 'women' and 'public' are nearly exclusive terms, yet women did worship in the public sacral space which was the church (Randolph 1997, p. 18). Women were restricted in various degrees from sacramental proximity (Randolph 1997, pp. 29, 39; Cooper 2011, pp. 92, 95, 100–1; Romagnoli 2016). They were, however, recipients of much pastoral care, particularly from the mendicant orders, not only for the purposes of instructing their children in the faith, but also due to the predominately female character of late medieval devotion and penitential piety. It is thanks to a number of works written or preached by mendicant friars that we know what they wished for in domestic devotion and when, and how, people, especially women, should pray. (Gagliardi 2013, pp.117, 125). Maria Pia Paoli has shown how spiritual treatises addressed to women, particularly those of Dominican tradition, fused with secular behavioural manuals and humanist works in promoting the role of the family as a unifying building block of civic life, with the family 'understood as a depositary of affective values' (il ruolo unificante della famiglia intesa come deposito di valori affettivi)' (Paoli, p. 1).

The confinement of women to the domestic space is evident in some of these works. In the Regola, Dominici warned Bartolomea degli Obizzi that women should obey their husbands and never stay overnight elsewhere, even on pilgrimage (Dominici 1860, pp. 87–88). If a husband forbids visiting relatives or friends, he should be obeyed; he should even be obeyed if he forbids going to church on ordinary days—the wife should pray at home instead. Only on Sundays and important feast days should women attempt to disobey, if they can, in order to attend church (Dominici 1860, pp. 90–91). He advocated caution to those who wish to live in penitence as mendicants, advising that there are many false followers of Dominic and Francis who sow discord rather than peace (Dominici 1860, pp. 98–99). Dominici expressed particular concern for a woman enclosed as a hermit, primarily because she would be unable to hear sermons and solitude can bring dangerous thoughts (Dominici 1860, p. 100).

As noted above, suspicion was voiced by Bernardino da Siena about women's motivations for going to church when he reprimanded women who only went to church to be seen, or worse, to show off their daughters to potential suitors (Bernardino da Siena 1888, III, pp. 212, 381). Caution for women was not only limited to the giving of alms, but also in going on pilgrimage. Pilgrimage was routinely cited as a space for potential danger. Dominici, as noted above, instructed Bartolomea degli Obizzi to never stay overnight outside the house, even on pilgrimage, although he concedes that husbands, too, should not remain away from their wives without their permission unless they are on a crusade (Dominici 1860, pp. 87–88). San Bernardino da Siena (1380–1444) echoed other preachers in fearing that more sin could be committed on pilgrimage than value gained (Bernardino da Siena 1884b, p. 380). Over a century earlier, the Dominican friar Giordano da Pisa (d.1310) preached in Florence that the Virgin should be a role model for women not because she fasted, or went on pilgrimages, except rarely little, or gave alms, or served in hospitals, but because she loved God above all things (Delcorno 1975, p. 72). The Vallombrosan monk Don Giovanni dalle Celle (d.1396) rebuked a religious woman, Domitilla, when he heard that she intended to go to the Holy Land, despite it having been urged by St Catherine of Siena, to whom he was otherwise devoted (Biscioni 1736, pp. 68–76).

The advice given by friars and clergy to women to focus their devotions within the home could not overcome the tensions between a liturgical, sacramental religion in which a consecrated altar was essential, and an interiorized affective piety which could be practiced within the home. As seen above, friars preached against women going on pilgrimages and attending mass devotions, yet the evidence of Margherita Bandini Datini (c. 1360–1423) indicated that such activities were widely participated in by women. Margherita writes of how she and a party of women attended the devotions and indulgences at S. Maria Primerana, and how she went in on behalf of Francesco (Datini 2012, p. 169). Donato Velluti (1313–1370) described the piety of his wife, Bice Covoni, who went to Rome at the time of the Jubilee (1350), leaving Donato to care for their two children. He later wrote, after her death: '[She] was small and not beautiful; but wise, good, pleasant, loving, well-behaved, and of every virtue full and perfect, and who worked to love and wish good for every person: and I praised her much, as she loved and desired me with all her heart. She was very good of her soul: and it is to be believed, that Our Lord Jesus Christ will have received her in his arms, making good and the best works, alms, and praying and visiting the church, and having had, in her illness, pardon from guilt and penance from bishop Agostino Tinacci (Velluti 1914, p. 290).[4] The evidence of Bandini Datini and Velluti shows the prescriptive, rather than descriptive, nature of the warnings of the clergy. In neither case do we have indications of the secular men involved (Francesco Datini, Donato Velluti) expressing disquiet about their wives' pilgrimages.

> *Predicatrici: 'Oh ladies, tomorrow I want to make you all preachers.' Women and devotional instruction.*

Bernardino da Siena explicitly called on the women listening to his sermons to be preachers—predicatrici: 'Oh ladies, tomorrow I want to make you all preachers.'[5] He thus invested them with a teaching and converting role but safely circumscribed within the walls of the home, limited to the family and, indeed, the relaying of *his* words (Bernardino da Siena 1884a, p. 59). This echoes Dominici's advice to Bartolomea degli Obizzi, when he told her that she should preach to them, saying to them things that are not harmful, and then teach them how to preach themselves (Dominici 1860, pp. 146–147).[6] This could suggest an extraordinary degree of trust on behalf of both Dominici and Bernardino, and, given clerical beliefs about the weakness and lack of intellectual capacity of women, can be explained only by the fact that it is they, the mendicants, who are telling the women what to preach, and that the women are, in effect, mediators passing the doctrine of the mendicants on to their children.. Orthodoxy could be checked by the dynamics between the domestic and the religious in the form of confession, and, it is to be noted, heretics could be discerned by their absence from confession and liturgical offices, exemplified by the trial of Giovanni da Montecatini, burned in 1450. The dangers that the home represented were shown when Giovanni was reported to have never been in seen in S. Lorenzo, his parish church, but to have declared that his own dwellling was a church where he received the Eucharist (Morçay 1914, p. 168). The home was the crucible of lay Christian formation and it was essential that its occupants were orthodox. Bernardino also, surprisingly, told women that their primary duty was to look after the household, and only after such duties were fulfilled, were they to go to the sermons. 'Is there someone who has a sick person at home? – Yes. – Do you not know how important his care his? Do not abandon him to come to the sermon. Have you children? – Yes. – Do not abandon he who has need of you, to come to the sermon. Have you husband and children, who need that they be looked after with the necessity of the family? – Yes. – Do, do it so

---

[4] 'La quale fu piccola e non bella; ma savia, buona, piacevole, amorevole, costumata, e d'ogni vertù piena e perfetta, e la quale si facea amare e volere bene a ogni persona: e io molto me n'ò lodare, chè me amava e disiderava con tutto quore. Era bonissima dell'anima sua: ed è da credere, che Nostro Signore Iesù Cristo l'abbia ricevuta nelle sue braccia, faccendo buone e ottime operazioni, limosiniera, e d'orare e visitare la chiesa, e avendo avuto, nella sua infermità, perdono di colpa e pena dal vescovo Agostino Tinacci.'

[5] 'O donne, domain vi voglio fare tutte predicatrici.'

[6] 'Insegna loro predicare poi hanno veduto alcuna volta predicare in chiesa, e tu predicherai a quegli dicendo cose non nocive e di sollazzo, e poi facendoli predicare stando tu con la tua famiglia a sedere basso quando in alto dicono, non ridendo ma commendando, e premiando quando ha contraffatto l'ufficio spirituale.'

you do not leave them to come to the sermon; do it that first you look after the house with those things that they need, and then, come to the sermon . . . ' (Bernardino da Siena 1884b, pp. 42–43).[7]

After speaking about how states should be governed, Bernardino connected civic governance to how women should govern the home, addressing himself directly to his female audience: 'And as I taught how men should rule the palazzo and the city; so also I want to teach women what they should do in their homes' (Bernardino da Siena 1884b, p. 41).[8] Bernardino even felt that this instruction was so important that if women had to choose between mass and the sermon, they should choose the sermon (Bernardino da Siena 1884a, p. 66).[9] To Bernardino, the parents' role was clear, they were to make the children like the angels of Paradise. 'It is necessary to teach children the Ave Maria, the Pater noster, the Credo in Deum, that every Christian is obliged to know; and leaving the house they should sign themselves with the sign of the holy cross. And when one gets up, and when one goes to bed; that they should say five Pater nosters and five Ave Mariaswhen one goes to bed, and one rises the Salve Regina, and other prayers . . . And all the said things the mother should teach them.' The father, instead, was to offer instruction involving other spaces: he was to teach them the hearing of mass, sermons, vespers, how to confess and then how to behave in the secular world and guard themselves against bad contracts, gambling, going to taverns, and bad company (Bernardino da Siena 1935, pp. 65–66).[10]

The emphasis on women's responsibilities can be seen again in the case of Margherita Bandini Datini. She was entrusted by the Dominican preacher Antonio Cancellieri with a book of the seven penitential psalms and a sheet with the prayer of St. Birgitta of Sweden, 'which she [Birgitta] made when the Body of Christ was raised' so that she could teach it to her little niece (Brambilla 2010, p. 19). Bandini Datini's letters talk of how Caterina (Tina) 'has read the Psalter; she needs a small prayer book that has the seven psalms and the Office of the Virgin with good lettering' (Datini 2012, p. 106). In the inventories researched by Christian Bec, some psalters are described as specifically for children, 'j saltero da fanciulli' (Bec 1984, p. 152). A century later, the wife of Lorenzo il Magnifico de' Medici, Clarice Orsini (1450–1488), had, to the great discomfort of the humanist Angelo Poliziano (1454–1494), undertaken the teaching of her young son, Giovanni, changing the reading prescribed by Poliziano to the Psalter: 'As for Giovanni, you will have seen for yourself. His mother has taken it upon herself to change his course of reading to the Psalter, a thing I did not approve of. While she was absent he had made wonderful progress. He was already able to select, without any help from me, all the letters and syllables in his exercise in composition' (Ross 1911, p. 216).

The domestic space was an ideal one to inculcate the piety of children with the use of fictive altars and dolls. In a famous passage cited in the introduction to this essay, Dominici urges Bartolomea degli Obizzi to instil in her children a love of the church by transposing the church into the home (Dominici 1860, pp. 130–133). The children are to dress as priests and to create little altars, decorate them, clean them, adorn them with images. Dominici was drawing on a long tradition. The miniature altars to be recreated in a domestic space are, according to Danièle Alexandre-Bidon found already in

---

[7]　'Ècci chi abbia lo infermo in casa? – Sì. – Non cognosci tu quanto bene fa il governo suo? Non l'abandonare per venire alla predica. Hai figliuoli? – Sì. – Non gli abandonare di quello che hanno bisogno, per venire alla predica. Hai il marito e' figliuoli, i quali bisogna che sieno governati di quello che bisogna alla famiglia? – Sì. – Fa', fa' che non gli lassi per venire alla predica; fa' che tu prima governi la casa di quelle cosec he bisognano, e poi viene alla predica [...]'

[8]　'E perchè io ho insegnato come debbano règgiare gli uomini il palazzo e la città; così anco voglio insegnare alle donne quello che debbano fare alle case loro.'

[9]　'E se di queste due cose tu non potessi fare altro che l'una, o udire la messa o udire la predica, tu debbi piuttosto lassare la mess ache la pedica; imperò chè la ragione ci è espressa, che non è tanto pericolo dell'anima tua a non udire la messa, quanto è a non udire la predica.'

[10]　'Dèbbesi a' fanciulli insegnare *l'Ave Maria*, il *Pater noster*, il *Credo in Deum*, che ogni cristiano è tenuto di saperlo; e che a l'uscita di casa si segni del segno della santa croce. E così quando si leva del letto e quando si corica; e che dica cinque *Pater nostri* e cinque *Ave Marie* quando si va a letto, e quando si lieva la *Salve Regina*, e dell'altre orazioni . . . E tutte le dette cose debbe loro insegnare la madre. Il padre: l'udire della messa, le prediche, il vespro, le laulde, insegnarli orare, confessare, e che si guardi da ma' contratti, e che non giuochi, e che non esca di casa la notte con bullettini o sanza. Ammunirlo che si guardi dalle taverne, dalle ghiottornie, e che digiuni e dì comandati, e che si guardi dalle cattive compagnie e dalle male usanze.'

the twelfth-century work of Guillaume of Auvergne (Alexandre-Bidon 1998, p. 1176). Later, Giovanni Certosino recommended that women should pray in front of a small altar decorated with images (Miglio 2008, p. 235). It was perhaps at altars like these that children were instructed on how to receive holy communion. Franco Sacchetti (d.1400) wrote of how his contemporaries taught children to hear confession and to take communion by giving them unconsecrated hosts (Sacchetti 1857, p. 29).

The domestic space could be sacralized through the possession of relics. Bernardino wrote of how some at home may hold the relics of the cord of St Francis, or bits of cloth from his cape, but, again reinforcing the sacrality of the ecclesial space, he reminded his audience that the most precious relic of all was the Body of Christ (Bernardino da Siena 1884a, pp. 237–38). It is interesting to note Bernardino's concern with relics, given their scant appearance in inventories before 1560 (Anderson 2007, p. 114). Women were believed to be more prone to superstition and thus needed to be protected against believing too quickly in unlikely relics (Dominici 1889, pp. 62–63). Bernardino, too, railed against too easy belief in relics such as the milk of the Virgin and the wood of the True Cross (Bernardino da Siena 1884b, p. 375).

## 4. Devotional Texts

As seen in the advice given by friars to women, spirituality was encouraged by looking at holy images and reading or hearing devotional texts (Paoli, pp. 17–19). The relationship between women and written culture has been explored by Miglio (2008, pp. 57–62) and Kaborycha (2012, pp. 793–95) who caution against believing in the inflated figures given by Giovanni Villani for female literacy but indicate that the circulation of texts among bourgeois Florentine women shows a familiarity with and interest in written culture, at least in vernacular written culture. Inventories, such as the ones studied by Bec (1984), reveal the titles of well-known devotional texts, but Florentines also kept miscellanies, or zibaldoni, similar to English commonplace books, in which we can find examples of sermons, doctrinal treatises, instruction manuals, legends of saints, prayers and other religious writings. The ubiquity of these miscellanies can be seen in their survival in Florentine libraries and the range of materials and social categories of owners has been studied by Gill (1994), Kent (2000), Miglio (2008), Bryce (2005, 2009), Kaborycha (2012), and Corbellini (2019). Dominici's Regola, although not believed to have been widely available, was copied out by Caterina di Donato Alderighi, wife of Zanobi di Taddeo Gaddi, son of the artist, in 1410 (Boehm and Kanter 1994, p. 33).

Books of hours, rarely as lavishly illustrated as Northern European examples, were found in the inventories, sometimes described as specifically women's, such as the 'libracio di donna', listed in Piero Beni's inventory of 1431, which belonged to his wife, Mona Sandra (Bec 1984, p. 175). Margherita Bandini Datini wanted her book covered with black satin, so that it would not resemble that of a nun: 'am not happy about sending you vermilion samite for my prayer book. I want black. If you want to fob me off with the nuns' style of cover, there is no need to tell me. I could do that myself. I don't want any sort of white or pink on it because it would dazzle me. I want it black. Please get it done quickly, because it upsets me that I am not reading the hours as I used to.' (Datini 2012, p. 227).

The Lives of the Desert Fathers of St Jerome, loosely translated by Fra Domenico Cavalca, showed the exemplary lives of the desert fathers and mothers, and was a particularly useful text for the penitential movement, dominated, as it was, by women. Delcorno has compiled an index of manuscripts owned by women (Delcorno 1977, p. 679). One of these manuscript owners was Costanza, widow of Benedetto Cicciaporci, whose inscription on the text of BNCF II, III, 89 (Magl. XXI, 123) noted that she had the book written and made for the consolation of her own soul and secondarily for that of her daughters (Gill 1994, p. 97). The book was inherited by one of these daughters, Lucrezia (Bryce 2005, p. 149). The codex also contained extracts from the *Fior di Virtù* and Petrarch's *Trionfi* (Bryce 2005, p. 149).

The affective devotional text, *Meditationes Vitae Christi*, written for a Clarissan nun and advocating the reader/listener to imagine themselves present at various moments of the life of Christ, narrated with immediacy and charged with emotional affect (Green and Ragusa 1961; McNamer 2009; Holly 2009) is

also found in codices owned by women, such as BNCF MS II, VIII, 25, containing the *Meditationes Vitae Christi*, a fragment of the life of St Elizabeth of Hungary, a passage from the gospel of St John and two laude, which at one time belonged to Antonia, wife of Daniele Canigiani (Mazzatinti 1901, p. 230). Antonia was the daughter of Tommaso di Zanobi Ginori, who married Canigiani in 1434 and died 30 December 1457 (Passerini 1876, Tav, II).

The lives of the saints were popular, such as one contained in MS Panciatichiano 39 which is inscribed at f.91r: 'Iste liber est domine chatarine, uxoris domini bonifacj de lupis, ut omnes qui habebunt hunc librum in suis manibus deprecor orent pro anima eius' (Morpurgo 1887, pp. 75–79). Its owner, according to the inscription, Caterina Franzesi, was the daughter of Antonio di Niccola dei Franzesi da Staggia and Cina di Francesco Cinughi. Widowed in 1389, she was the executor of the testament of her husband, the mercenary originally from Parma and formerly at the service of the Carrara of Padua, Bonifazio Lupi, which, among other things, established the hospital known as the Spedale di Bonifazio (Angiolini 2006). She maintained a supervisory role over the construction and decoration of the hospital. She was also a Franciscan tertiary (Passerini 1876, p. 226).

Petrucci has shown that these vernacular texts, copied and recopied by the laity, were kept with letters, account books and family documents in chests and read in the home or the shop, and that they 'belonged to the sphere of leisure and free time'; a sphere, that is, that was not ecclesiastically circumscribed (Petrucci 1984, pp. 612–13). Letters were frequently shown to others and copied. The notary Ser Lapo Mazzi, in his letter to Francesco di Marco Datini, refers to how he will show a letter from the Dominican prioress Chiara Gambacorta to Guido di M. Tommaso del Palagio (c. 1335–1399) when he has time: 'When I will have time, I will show the letter of the Sister to Guido' (Mazzei 1880, I, p. 116).

## 5. Devotional Images

Numerous small panel paintings survive from late medieval Italy, likened in their popularity by Victor Schmidt to books of hours in northern Europe (Schmidt 2005, p. 9). The function of these paintings was supposedly to help prayer and meditation and, thus, they did indeed serve a similar function to books of hours, themselves described as 'painted prayers' (Wieck 1997). Dominici did not hesitate to recommend the type of works that Bartolomea degli Obizzi should keep at home. In a frequently cited passage, he said that she could have images of the Virgin and Child, and the Massacre of the Innocents in order to teach little boys to shun violence. For little girls he wanted to see images of SS Agnes, Cecilia, Elizabeth of Hungary, Catherine of Alexandria, Ursula, and other figures that would give them a love of virginity, a hatred of sins, and a desire to flee bad company (Dominici 1860, pp. 130–31). Dominici's advice regarding the Massacre of the Innocents seems to have gained little traction, judging from the subject matter of surviving Florentine panels, but the saints he recommended do frequently appear (Lawless 2011, pp. 235–37). Dominici did not just expound on subject matter, but also on the type of painting that should be kept in the home. He did not like the use of silver, gold or precious stones, fearing that the young would become idolatrous and love the materials rather than the representations of truth depicted within the image; the older and more smoke-blackened through the maintenance of candles, the better (Dominici 1860, pp. 132–3; Freedberg 1989, pp. 11–12).

As Wilkins has pointed out, the survival of so many trecento tabernacles indicates widespread use, but little evidence suggests how or where they were used. Some, certainly, must have been used in religious houses in the private cells of the religious. He cites the important observation of Tommaso da Celano in his *Vita Prima* of St Francis, wherein he spoke of how Roman wives had a painted icon in their houses on which their favourite saint was painted (Wilkins 2002, pp. 371–72). This evidence is important, as it connects images of saints with women and domestic devotion, but Celano is clear that it is a custom of Roman women and we cannot be certain that the evidence can be related to Umbria and Tuscany. Anderson has demonstrated that exempla and sermons suggest that *altarini* were common, but that there is little other evidence to prove it (Anderson 2007, p. 37). Yet we know

from the inventories that holy images existed in homes. Jacopo di Filippo Guidetti's inventory of 1427 included a small panel painting of the Crucifixion (Musacchio 2008, p. 198). Crisi elaborates that, in his house, he had a panel of the Virgin in the bed chamber of the ground floor, another panel 'quasi nuova' in his chamber, and another 'all'anticha' in the chamber over the garden. Further, he had a panel with 'Santa Ilaria' and a bone tablet with the Crucifixion (Cristi 2012, p. 873). In his villa at Vingone, he had three painted panels, one of St Jerome, one of the Crucifixion, and the third 'dipinta alla crecha, vecchia' (Cristi 2012, p. 874). Giovanni Boccaccio's 1374 testament left, to Sandra di Giovanni Sassetti, a little panel, on which was painted the Virgin Mary with the Child and, on the other side, a skeleton's skull (Lami 1758, I, p. 137).

The inventories show, thus, images in the home of the Virgin and Child, the Crucifixion, and various saints. Wilkins suggests the standardized iconography with the dominance of the Virgin and Child suggests an audience of women (Wilkins 2002, p. 372). Given the injunctions of the clergy on women as guardians of the faith within the home, the likelihood of tabernacles serving as the type of 'little altar' envisaged by Dominici is strong. The continuity of religious imagery being purchased or commissioned by private patrons in the Quattrocento can be seen in the *ricordanze* of the painter Neri di Bicci, many of whose clients were women (Gilday 2001, pp. 51–75).

Some evidence of how such images may have functioned has been gathered by Victor Schmidt in looking at depictions of St Catherine of Alexandria and the hermit. In this late medieval legend, Catherine is introduced to her future spouse by a hermit, who shows her a panel of the Virgin and Child. When the Virgin and Child appear to her in her chamber, she recognizes them as the same figures as in the panel shown to her by the hermit. The altarpiece showing the life of St Catherine by Donato and Gregorio d'Arezzo shows Catherine's mother taking the panel from the hermit, and shows it as clearly portable (Donato and Gregorio of Arezzo, *St Catherine of Alexandria Vita panel*, 100 × 172 cm, Malibu, J. Paul Getty Museum (Schmidt 2005, pp. 9–26).

Further evidence on how such a panel may have been used is shown in the life of the Florentine saint Umiliana dei Cerchi (d.1246). Umiliana is a useful example as, although she was promoted by the Franciscans immediately after death as a saint. Forbidden by her family to enter a religious order on becoming a widow, she enclosed herself within a tower of the Cerchi palazzo. Having refused remarriage, she dedicated herself to a life of prayer and charity, but some references to her children indicate that they remained with her rather than with her marital kin. We learn that Umiliana prayed in front of a painted image of the Virgin Mary to decide on her future, and then resolved that she would rather be burned in a furnace than remarry (De Luca 1977, III, pp. 370–371); that she kept a lamp burning in front of the image, as when it blew out a white dove came into the room, illuminating it by means of a vermilion rose in its mouth; another time when the lamp went out it was relit by the hand of an angel (De Luca 1977, III, p. 376). Another miracle recounted by the hagiographer was when her little daughter fell out of a window and lay on the ground as if dead. Umiliana brought the image of the Virgin and Child to the injured child and prayed, after which the Christ Child himself came out of the picture, blessed Umiliana's daughter, and she got up, completely cured (De Luca 1977, III, pp. 395–96). At her death, her companion brought her the image and we learn then that there was a precious relic inside in the form of a lock of hair of the Virgin. She lit candles and formed them in the shape of a cross and placed them with the image on Umiliana's breast, lit incense and sprinkled holy water on her head and Umiliana then, after rejecting the devil, placed the image in the silk cloth of her mantle and 'positioned it better upon her breast' (Schuchman 2009, p. 383; De Luca 1977, III, pp. 401–2). The vita shows us that the image of the Virgin was the chief focal point of the prayers and devotions of Umiliana, that it was portable, and when laid on her breast it provided comfort when she died. This can be compared with the devotion of the dying son of Giovanni Morelli in 1406, Alberto, who asked on his deathbed that he be brought the image of the Virgin and Child (Branca 1986, p. 294; Bailey 2009, p. 984). We also learn that the portability of the image lent itself to being touched and, thus, its importance was far from confined to the visual.

Typical of many devotional panels is a small panel, now in Boston, from the workshop of Bernardo Daddi (d.1348), depicting the Crucifixion with mourners and a kneeling female donor portrait figure (Figure 1). I intend the word 'portrait' here to indicate not a naturalistic portrayal of an individual (although it may be that), but rather, as Victor Schmidt uses it, as a surrogate, performative self, engaged in eternal prayer (Schmidt 2005, p. 117). In the panel, to the left, the fainting Virgin is supported by the holy women and the Magdalen embraces the cross. To the right, the donor kneels, facing the cross and the Magdalen, her hands joined in prayer, while Longinus gestures to the cross. The crucified Christ occupies the top half of the panel, with mourning angels catching his blood. The panel is perfectly suited to meditation of the kind advocated by Dominici and others in domestic spaces. The donor figure herself, perhaps, could merge with her effigy in praying and contemplating the passion and body of Christ, or her family could pray in front of the panel and remember her soul in their prayers, thus joining the worlds of the living and the dead. The panel reminds the viewer of the importance of the sacrifice of Christ and the Eucharist, through the blood being caught in chalices by the angels. The penitent state necessary to receive the Eucharist is evoked by the Magdalen and the sorrows of the Virgin echo the words powerfully wrought in texts such as the Meditationes Vitae Christi.

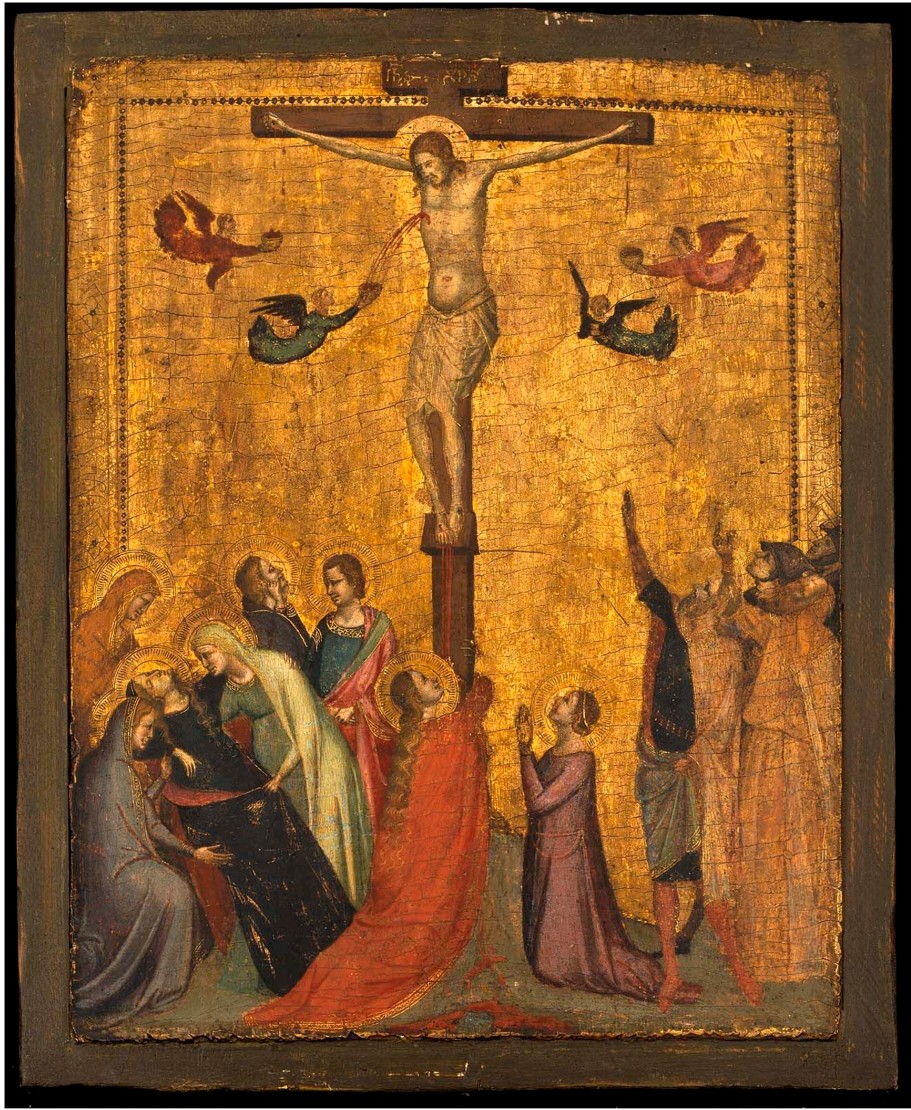

**Figure 1.** (Close follower of) Bernardo Daddi, *Crucifixion with kneeling donor*, c.1320, wooden panel, probably part of a diptych, 40 × 28.5 cm. Boston, Museum of Fine Arts, 23.11.

The workshops of artists like Bernardo Daddi and Jacopo del Casentino produced vast numbers of small panels, diptychs and triptychs for personal and domestic devotion. The popularity of saints such as Bernard, whose writings are found throughout the miscellanies owned by the devout, is echoed in the iconography of small-scale devotional works, such as the so-called Cagnola triptych, (Figure 2) the only signed work of Jacopo del Casentino (d.1349), where St Bernard is seen in the central panel with St John the Baptist flanking the Virgin Mary. On the side panel is the Stigmatization of St Francis with two virgin martyrs (probably Lucy and Margaret) and, on the other side, the Crucifixion of Christ. The entire triptych measures only 39.2 × 42.4 cm (http://www.polomuseale.firenze.it/catalogo/scheda.asp, no.00282237).

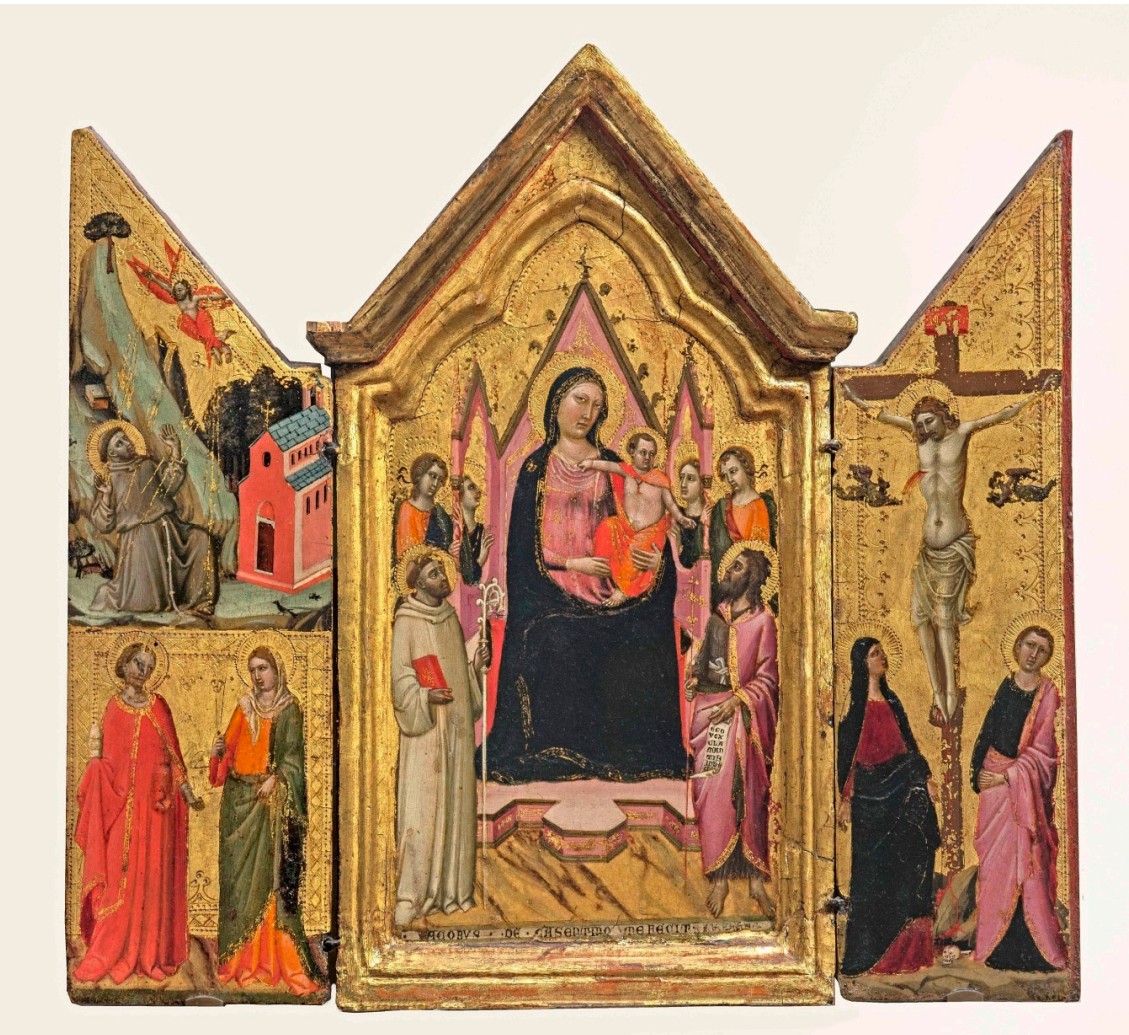

**Figure 2.** Jacopo del Casentino, The 'Cagnola' triptych: Virgin and Child with angels and Saints Bernard, John the Baptist; Stigmatization of St Francis; SS Lucy and Margaret; Crucifixion, panel, 39.2 × 42.4 cm, c.1320–1349. Florence, Uffizi, 9258.

It is, of course, impossible to know how images were 'read'. We rarely, if ever, have direct testimony of the relationship between the viewer and the viewed, but the work of Antoninus Pierozzi, in his advice to the widow Dianora Tornabuoni, although nearly a century later, could provide some useful clues, given the consistency of Dominican spirituality and instruction to pious women. The stigmatization of St Francis shows the moment in which Francis, rapt in prayer in La Verna, was imprinted with the marks of the Passion of Christ and became an alter christus, perfected through prayer, and naturally links with the Crucifixion on the other wing, a scene which devotional texts

urge their readers and listeners to meditate upon and to move 'with the eyes of the mind' over the bleeding wounds of Christ (Pierozzi 1858, pp. 169–70; Flanigan 2014, p. 181, Lawless 2014–2015, p. 64). The angels catching the blood of Christ remind the viewer of the Eucharistic significance of the body of Christ, with the viewer, adequately prepared after confession and penitence, being encouraged to receive Communion. Tornabuoni, whether she was at home or at church, was to gaze at the crucifix and move across the body of Christ with the 'eyes of the mind', saying Ave Marias and Paternosters, dwelling on individual details such as his blood, his sputum-flecked beard, his forehead and, most particularly, his wounds (Pierozzi 1858, pp. 169–70). The presence of St Bernard, reputed author of *De Planctu Beatae Mariae*, or the lament of the Virgin on the Passion of Christ—a well-known text in Florentine miscellanies—could also remind the viewer of the future sacrifice of the Child he gazes at (Mâle 1986, p. 468). It could be hypothesized, with the help of Antoninus, that such a triptych could help prayer in a complex interplay of image, prayer, texts—remembered or read—and the devotee or devotees. The distinct spaces of the triptych could easily, if required, be used for different devotional exercises and compartmentalized, with the viewer moving from one image to another to culminate, perhaps, in the Virgin and Child or the Crucifixion. The Child, as in so many of these small panels, appears playful and active, reaching out to the angel or saint at his side. The saints gaze up at the Virgin, while she serves as visual interlocutor to the viewers, inviting them to join the saints in prayer.

Another small triptych, now in the Lord Methuen collection in Corsham Court (Wilts.) (Figure 3), painted by Puccio di Simone, is explicitly linked to female devotion by the presence of a kneeling female figure (Fremantle 1975, p. 92). The central panel shows the Virgin and a standing, playful child touching her chin and holding a chaffinch. Angels stand on either side of them, with their arms folded, and below is St Margaret and St John the Evangelist. In the side wings are four scenes, on the right, the Lamentation over the Dead Christ; the Mystic Marriage of St Catherine of Alexandria; the Noli Me Tangere; and the Feeding of SS Anthony and Paul in the Wilderness. It is in this latter scene that the female figure appears. Her flowing hair indicates that she is not married, and her richly lined dress, with a square-cut neck, shows that she is not a member of a religious order. She is not, thus, yet a mother or wife with a family to instruct in religious devotion but could be a young woman on the cusp of marriage, with a panel containing suitable themes for a Christian wife or for a young woman about to enter a convent. The central panel's Virgin and Christ Child is, of course, a maternal image, which is enhanced by the presence of St Margaret, patron saint of childbirth (Musacchio 1999, p. 143) and frequently represented in Florentine small-scale panels (Lawless 2011, pp. 254–55). St John the Evangelist was the most beloved of Christ, the author of the Apocalypse, and was related to Christ, being his first cousin according to the apocryphal accounts used in many Florentine miscellanies (Lawless 2010). The Lamentation scene connects directly to the Noli Me Tangere through the presence of the Magdalen. The penitence personified by the Magdalen then connects to the penitential and austere lives of SS Anthony and Paul in the desert, and the need to expiate sin by appropriate fasting and oration. St Anthony's life was popularized in Cavalca's *Vite dei Santi Padri* (Cavalca 1858, pp. 17–38). The association with food and fasting is particularly associated with female penitence, as shown by Walker Bynum, and is a key component in the lives of many late medieval female saints (Bynum 1987). Margherita Datini records a contemporary of hers, Maria Strozzi, a widow, whose abstinence was so severe that 'Her mother and father believed she had been sleeping only on a board and they thought she had been fasting so much that she had consumption.' (Datini 2012, p. 57). Giordano da Pisa, preaching on the feast of St Mary Magdalen, 22 July 1305, told his audience how women could be among the the greatest saints, using the example of the Magdalen, who, as she had been rewarded with angelic food exceeded other hermits such as Mary of Egypt and Paul the Hermit whose sustenance consisted of herbs (Moreni 1831, I, p. 182).

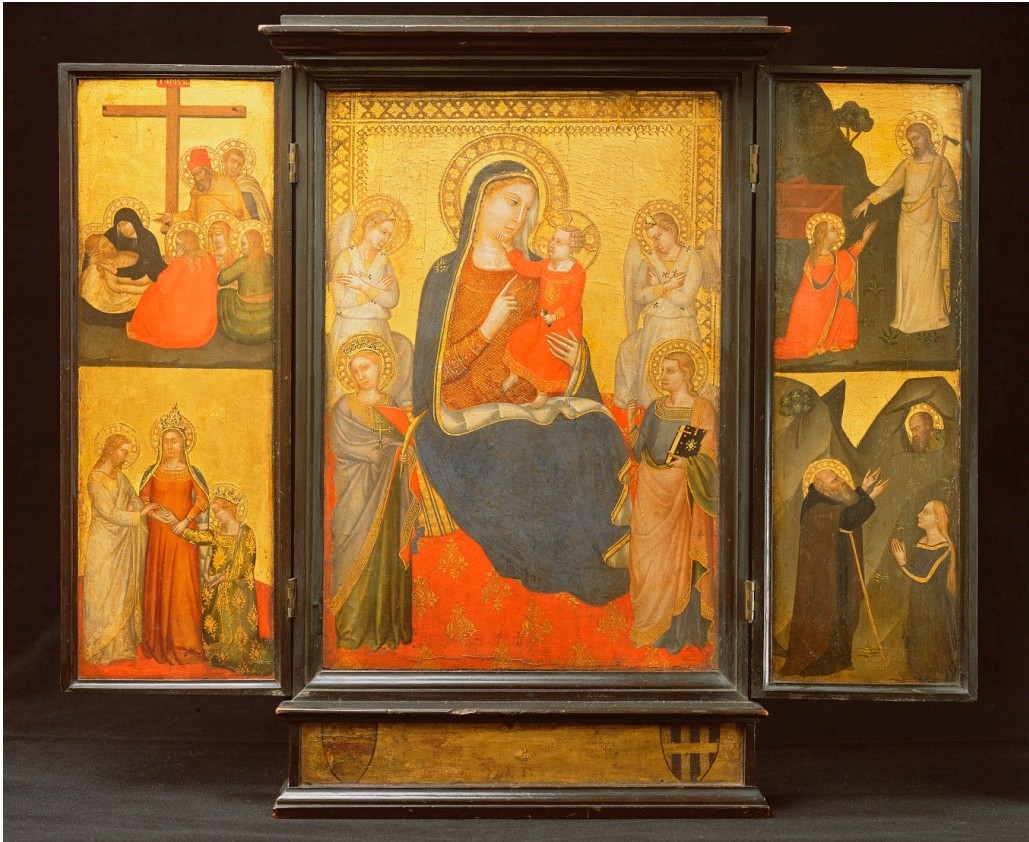

**Figure 3.** Puccio di Simone, Virgin and Child with angels and SS Margaret, John Evangelist; Lamentation over the Dead Christ; Mystic Marriage of St Catherine of Alexandria; Noli Me Tangere; St Anthony Abbot and St Paul the Hermit with kneeling donor, c.1350, panel, 39.5 × 26.0 cm (centre), 39.5 × 12.9 cm (each wing). Corsham Court (Wilts.) Lord Methuen Collection. Image courtesy of Corsham Court, Wiltshire/ Bridgeman Images.

The Mystic Marriage of St Catherine was a popular late medieval legend in which the virgin princess, Catherine, is rewarded by the Christ Child in a vision by the Virgin Mary joining him to her in marriage. Although there was a long tradition of mystic marriage within Christianity and Judaism representing a symbolic pact, perhaps between the soul and God, or the Church and Christ (Matter 1999, p. 31) the mystic marriage of St Catherine knew a particular fortune of its own, appearing first in visual imagery and only then in texts (Walsh 2007, p. 4). Schmidt points out the importance of the soul being joined to Christ as his bride in the contemplative life, and that, although married women led an active life, Giordano da Pisa had emphasised how a room or a cell could be transformed into a desert in acts of devotion (Schmidt 2001, pp. 24, 26). St Catherine is here presented by the Virgin to an adult Christ—a more destabilizing, yet popular, iconography which undermines both the role of male authority in sanctioning a union and, by placing the protagonists of the union as nubile, brings the mind more closely to marriage than the desexualised imagery of the Christ Child and St Catherine. Another possibility to be considered regarding the identity of the young woman is that she is not about to be married, but has died, and that the panel serves as a memorial and and invocation to her family to pray for her.

[This is part of the main text, not caption for the image.Main text begins here] Family devotion is suggested by the 1333 winged triptych of Bernardo Daddi now in the Museo del Bigallo in Florence (Figure 4) In the central compartment, the Virgin and Child are surrounded by fourteen male saints, and flanked by a male and female donor. On the inside left wing is the Nativity, and on the right, the Crucifixion with St Francis. In the spandrels is found the narrative of St Nicholas rescuing and restoring the little Adeodatus to his parents. St Nicholas was a particularly child-friendly saint, not only due to

his rescuing of Adeodatus but also in his provision of dowries to three poor girls and his resurrection of three young boys who had been pickled in brine (De Voragine 1993, I, pp. 21–27). In a mercantile city whose wealth was based on foreign trade, his role as a patron against storms at sea and shipwrecks was, doubtless, also important. On the outside of the wings is found, once again, the patron saint of childbirth, Margaret, with Catherine, Martin, and Christopher. Catherine was one of the most popular saints of the middle ages, partly due to her mystic marriage, which made her the daughter-in-law of the Virgin Mary, and also for the reasons enumerated by the Dominican Jacobus de Voragine (Jacopo da Varazze) in his Golden Legend, c.1260: 'Lastly, she was admirable by reason of her privileged dignity. Some saints have received special privileges at the time of death - for instance, a visitation by Christ (Saint John the Evangelist), an outflow of oil (Saint Nicholas), an effusion of milk (Saint Paul), the preparation of a sepulcher (Saint Clement), and the hearing of petitions (Saint Margaret of Antioch, when she prayed for those who would honour her memory). Saint Catherine's legend shows that all these privileges were hers." (De Voragine 1993, II, p. 341). St Martin's generosity in giving his cloak to a beggar was a perfect example of alms-giving and charity (De Voragine 1993, II, pp. 292–301). Another perfect family saint was the giant, St Christopher, a ferryman who carried people across a river and who unknowingly carried the Christ Child, weighed down with the sins of the world (De Voragine 1993, II, pp. 10–15); He was patron saint of travelers and perhaps his most important role was that of a protector against sudden death. It was believed that whoever looked on his image was safe from sudden death for that day (Rigaux 1996, pp. 241–43). It is, perhaps, telling that the only saint to feature in the frescoes in the house of the merchant Francesco di Marco Datini was St Christopher (Schmidt 2005, p. 100; Cole 1967, p. 80).

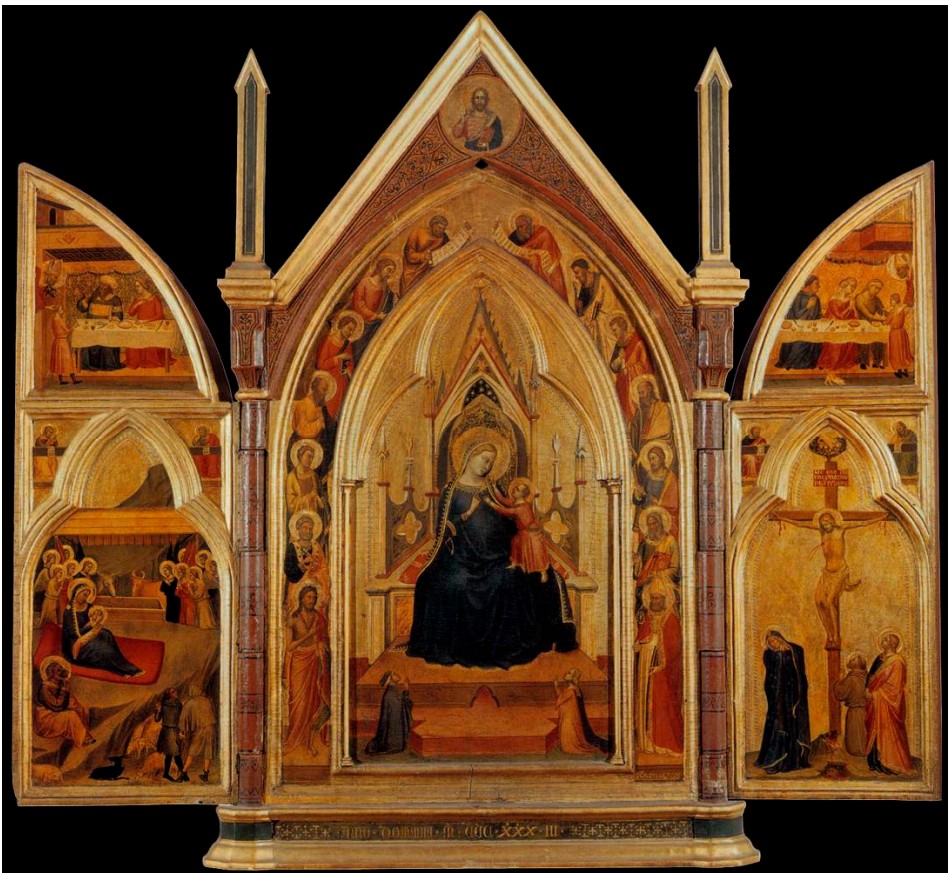

**Figure 4.** Bernardo Daddi, The Bigallo triptych. Virgin and Child with Saints; Nativity, St Nicholas rescuing Adeotatus from the pagan king; Crucifixion; St Nicholas restoring Adeotaus to his family, 1333, panel, 89 × 97 cm (closed). Florence, Uf.

The folding triptych in the Walters Art Gallery, Baltimore, also shows male and female donor figures (Figure 5). The central panel shows the Coronation of the Virgin with, below the Virgin, St Anthony presenting a male donor and, on the other side, below Christ, a male bearded saint (probably St Christopher) presenting a female donor figure. On the side panels are SS Nicholas and Peter, Paul and Catherine of Alexandria. Anthony Abbot was popularly invoked against not only St Anthony's fire, but a wide range of other illnesses and, in particular, the plague. On the exterior sides of the panel are SS Anthony Abbot and Christopher, with inscriptions indicating their patronal functions: 'Lucerna veri luminis, doctor humilitatis, magister et viator mirae sinplicatatis; refulsit in Egitto' ('A lamp of the true light, a teacher of humility, a master and traveller of admirable moral simplicity; he shone in Egypt') and 'Xpofori sancti spetiem quicumque tuetur, illo namque die nullo languore tenetur' (Whoever looks at the image of St Christopher will not be taken by any illness during that day') (Figure 6) (Schmidt 2005, p. 98).

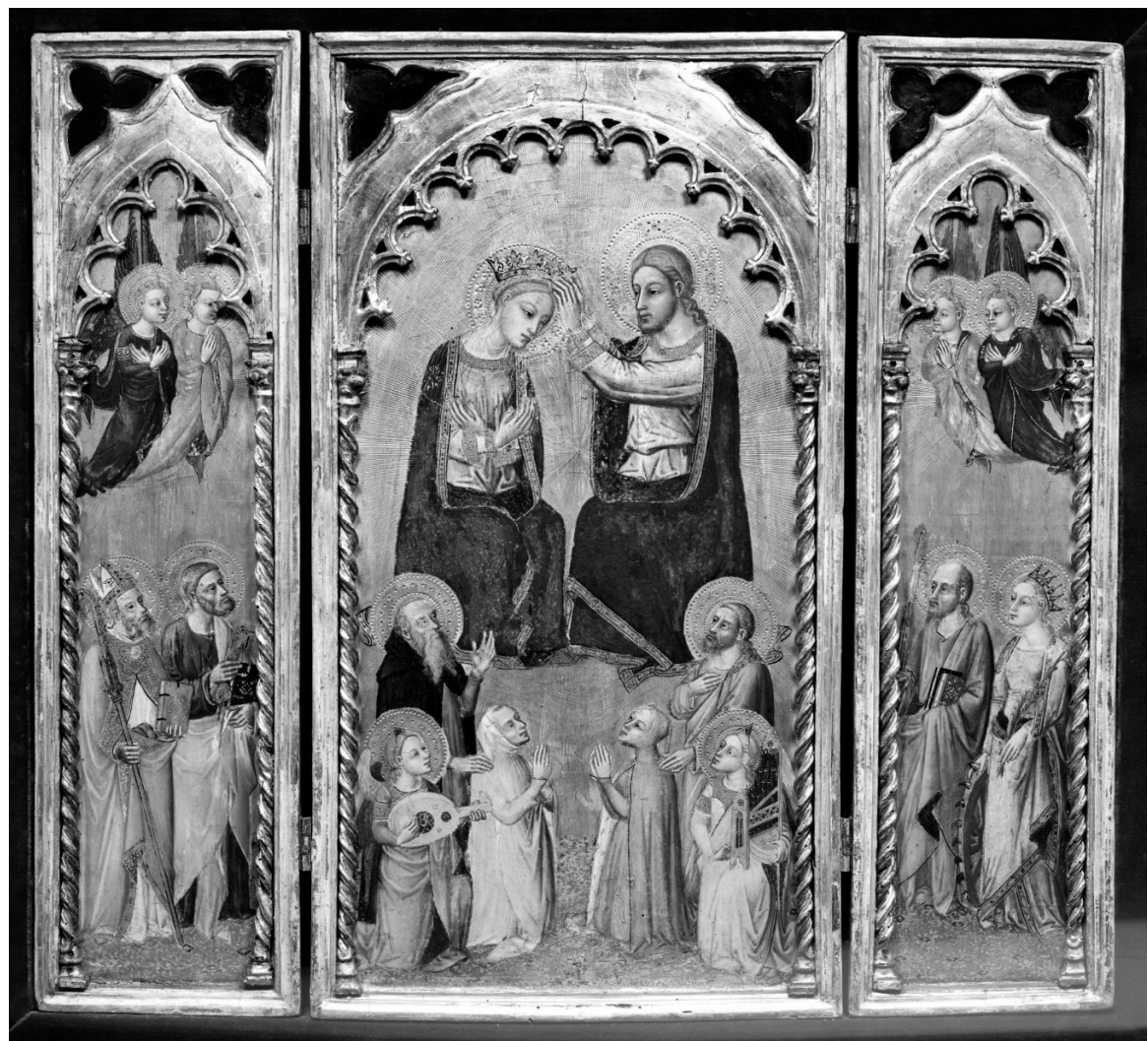

**Figure 5.** Niccolò di Tommaso, *Coronation of the Virgin with donors and saints*, c.1370, panel, 66.8 × 36.3 cm (centre), 66.8 × 18.1 cm (each wing). Baltimore, Walters Art Gallery, no. 718.

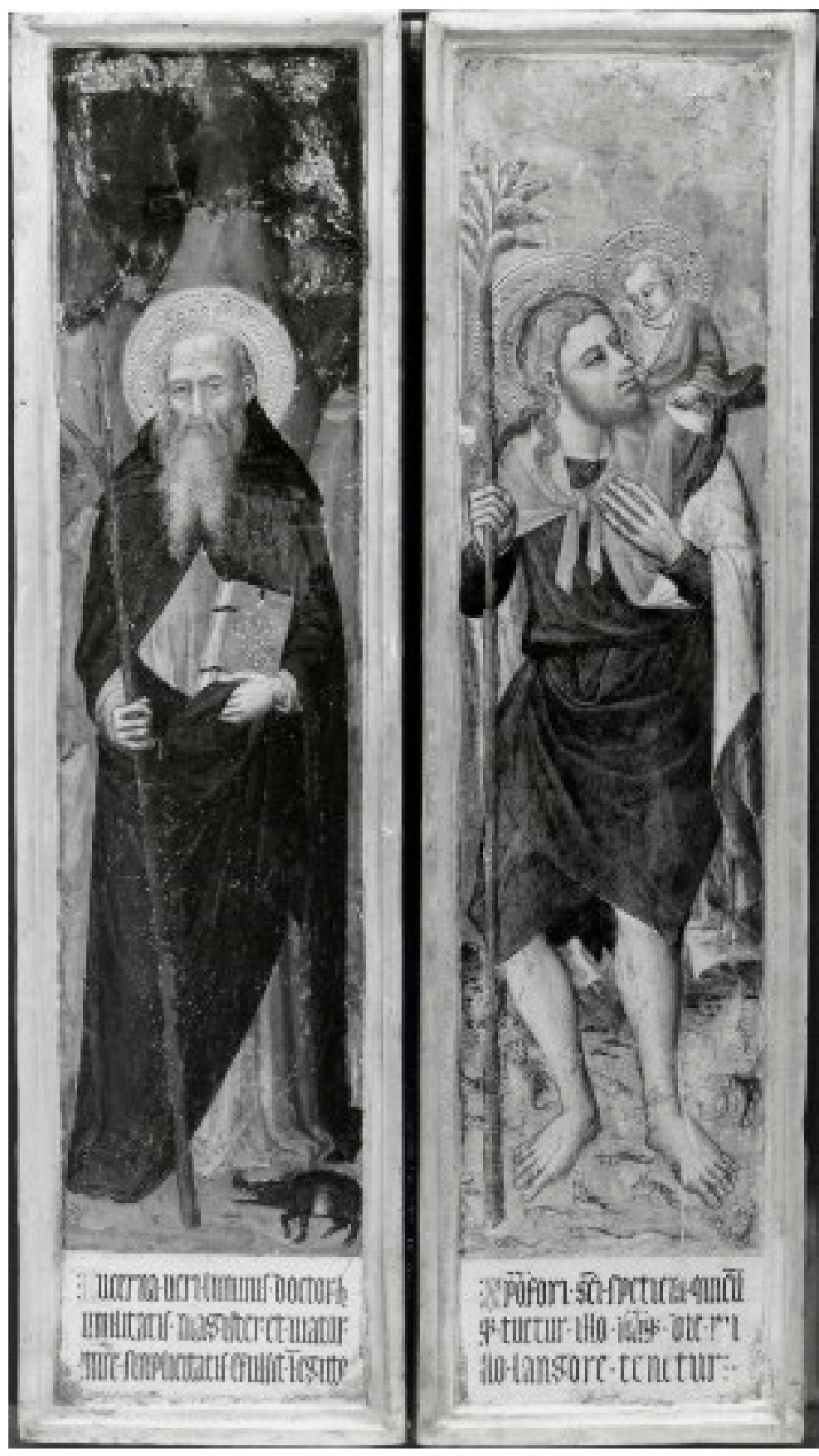

**Figure 6.** Niccolò di Tommaso, exterior shutters of wings of Coronation of the Virgin.

## 6. Conclusions

Panels like these were only a small part of the domestic, devotional culture of medieval Florence, and by and large little documentation survives on their patronage or early provenance. Later in the fifteenth century, more evidence exists on the art of domestic devotional milieu for families such as the Tornabuoni (DePrano 2018) and the Medici (Gombrich 1966), as well as the rich workshop records of Neri di Bicci (1419–1492) (Neri di Bicci 1976; Thomas 1995). The mendicants advised women how to use the crucifix in their prayer. Bartolomea was encourated by Dominici to embrace the crucifix repeatedly, to bathe it with her tears and to use it as tool for meditation (Dominici 1860, p. 61). Although these panels were costly and could not have been owned by many, even the poorest could own a crucifix. The Florentine apothecary and diarist, Luca Landucci describes how, in 1473, a poor woman, 'poveretta', recommending her many children to the crucifix in her home for prayer, suddenly saw it begin to sweat. Crowds gathered, and the Carmelite friars removed the crucifix and placed it in a tabernacle in their own church, transferring it thus from domestic to sacred space (Landucci 1883, p. 13) and, in so doing, reinforcing the hierarchy of sacrality.

This essay shows that male religious figures of authority were resolute in their advice as to where and how women could practice their faith. Their treatises directed women on the management of domestic space and the instruction of children. The ecclesiastical space was managed and curated in direct and considered opposition to that of the domestic realm, where women were (ideally) restricted to certain practices and had very different responsibilities to their husbands. They were kept distinct and isolated from the holiest of spaces and, yet, were expected to fulfil the necessary guidance and instruction of their children in religious instruction. The evidence of the treatises shows us the advice directed to, in particular, pious married women such as Bartolomea Obizzi, but the devotional literature read by women, and panels containing female donor figures indicate a broader basis on which to investigate religious devotions. The caustic letters of Margherita Datini, like those of Alessandra Strozzi a generation later, show a more pragmatic and mercantile approach to religious practice and merit further exploration in the field of female emotion and mentality. Finally, both inventories and the wealth of surviving trecento and early quattrocento small tabernacles and panels show how some religious themes and devotions were given a visual form. Some of these show indications of female patronage in the form of a donor figure or a donor couple, in the latter case, almost certainly linking the panel to the domestic space. Women, both religious and lay, could be said to have had a particularized agency within the domestic sphere, an agency which was negotiated with the help of mendicant friars. This guidance was underlined in texts, images and devotional practices in a cultural milieu in which the social fabric upon which the city depended was woven through the domestic spaces of the *ordo congiugatorum*.

**Funding:** This research received no external funding.

**Conflicts of Interest:** The author declares no conflict of interest.

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
