# Peer review of "‘Make Your House like a Temple’: Gender, Space and Domestic Devotion in Medieval Florence"

_religions, doi:10.3390/rel11030120_

Round 1

Reviewer 1 Report

The manuscript diligently provides a list of textual and art-historical references, but does not break a great deal of new ground in interpretation.  Indeed the abstract simply pledges to "discuss" domestic devotions.  While the source material is more than adequate for the task, the author's thesis is underwhelming.  Surely the author, having gathered a plethora of primary sources from the period in question, is able to be bolder in making constructive historical claims, even if tentative, about the relationship between public and private female religious space. 

The essay has a number of typos and the documentation is not standardized.  For example, short quotations are sometimes translated in the text and at other times relegated to footnotes.  At least one source cited (Shuchman) is not found in the reference list. 

Regarding the issue of relics, the Tridentine reforms of the 16th century raised the issue of authenticity and thus necessitated more stringent documentation.

This is a publishable piece, but only with revisions that sharpen the conclusion and correct for syntax and formatting.

Author Response

Response to reviewer 3

Thank you for the review, and firstly, may I apologize for the typos and lack of standardization in the documentation.  Another reviewer wanted me to provide translations in footnotes for the lengthier quotations – perhaps the editor will advise as to whether all, some or none should have the original language in footnotes. Schuchman has been added and I hope there are no other absences in the sources.

I hope that I have strengthened and sharpened my argument and used my evidence to make bolder statements about what I believe was a tense, but necessary relationship between the mendicants and women, the latter as representatives and leaders of popular devotions. I have amended the abstract accordingly. As my piece is concerned with the fourteenth and early fifteenth centuries, I did not refer to the Council of Trent and am unsure of how or where to put its decisions on the authentication of relics.

Reviewer 2 Report

I very much enjoyed, and was much edified by, the essay, "'Make your house like a temple': Gender, space and domestic devotion in medieval Florence."

I offer a couple of observations, coming from engagement rather than critique. The first thing that struck me was the "wildcatting" quality of the mendicant suggestions--precisely the sort of thing that gave Ordinaries pause when they considered allowing them to minister in their dioceses. In particular, I was taken aback by Bernadino's counsel that it was more important for his female hearers (his "predicatrici") to listen to his sermon rather than attend Mass! (L 271) More generally this prompted me to wonder whether the author has any evidence that allows her to situate her analysis in the familiar conflict between bishop and mendicant. I know from my own work that certainly from 4th Lateran on into the latter part of the sixteenth century, bishops were at least given the authority to regulate preaching in their dioceses and by the 1570s expected to be diligent in doing so. The sorts of devotions proposed by these mendicants for lay women seem at least as risky as unlicensed preaching, the more so since the female listeners are actually called "female preachers"! 

The same dangerous edginess--dangerous for the good order of dioceses at least--is suggested in Dominici's advice to Bartolomea that she preach to her family 'things that are not harmful,' a remarkably trusting assumption on his part. What standards of doctrinal verification are imagined? (Ll 256-257) So, what did the bishops say?

The author's punctiliousness about setting the proposed devotional practices in the context of class and gender prompted me to wonder how restricted the kind of practices she identified were, since the female hearers of Bernadino's sermons must have been very socially diverse, given his popularity. Any other evidence of the "lower order" (the author mentions "lower mental levels" at line 21) taking him up on his exhortations?

I found the observation that vernacular devotional books belonging to lay people were free from ecclesiastical circumscription precisely because they were not in the sacral sphere thought-provoking. When Visitor-bishops did their duty, they had the prerogative and the responsibility to rifle through the books and chests of their clergy to be clear that they had the books they were supposed to have and didn't have the ones they were not supposed to have--an interesting inversion.

The invitation to laywoman Dianora Tornabuoni to consider the bleeding wounds of Christ with "the eyes of the mind" (Ll 523-525) strikes me as risky even for mendicants, at least from the point of view of "good order," given the clear association of such "viewing" with suspicious (and hard to police) female mystical experiences. Taken together all of these examples suggest to me a surprising openness--at least on the part of the author's mendicant exhorters--to rethink (at least rhetorically) some of the possible roles in lived evangelism for women.  

I thought the close readings of the images which comprise the final section of the essay were particularly nicely executed (especially the 3rd figure), concretizing in a most thorough fashion the sorts of lay devotions (and the assumptions governing them) that the essay had dwelt upon. In that regard, since the essay concludes with such sustained and effective iconographical analysis, permit me to suggest that the author foreshadow that coming analysis even more than now in the excellent first section of the essay. 

I'll end by saying that the essay is embedded very effectively in a obviously well-read and well-understood secondary literature and makes an important contribution to lay--and especially lay female--devotional opportunities in the later Middle Ages and Renaissance.

Author Response

Response to reviewer 1

Thank you so much for your very encouraging review, which has helped me considerably and I think it has also provided me with new research questions on the relationships between these friars and their bishops. I was very surprised by Bernardino’s sermons, and have found little in the literature about him to help me explain such extraordinary advice.  Bernardino was investigated for heresy, but regarding his preaching of the Holy Name, rather than his advice to women. I think in both Bernardino’s and Dominici’s advice they intended that their doctrine, and only their doctrine, be communicated by the women within the home, rather than by giving them latitude to really preach or investigate doctrinal issues themselves. Bartolomea degli Obizzi and Bernardino’s listeners were the intermediaries, transmitting the mendicant message to their families. I think the invitation to be predicatrici an extraordinary one, especially as, unlike Dominici, he was preaching to a crowd rather than to an individual that he could have known and trusted. The views of Bernardino, Dominici and Antoninus found elsewhere are fully reflective of the traditionally negative clerical views on women, and I think that their invitations to these women are less to do with an expectation of any autonomous evangelism or spirituality, but rather a trust that they, as recipients of this expert mendicant advice, will pass it on. The case of Dianora Tornabuoni would support this, in that Antoninus gives such specific advice on exactly how she is to view the wounds of Christ. His relationship with her would mean that if she were to have a suspect mystical experience, he would be able to investigate it and if necessary, discipline her. Your review has given me a great deal to think about and I need to investigate much more the relationship between bishops and mendicants in Italy.

Reviewer 3 Report

The author of this article clearly has a good knowledge of primary sources and secondary literature. The amount of information gathered from primary sources is commendable, although some of it has been discussed by other authors, particularly Dominici and Bernardino da Siena.

However, within the context of the article, the knowledge of secondary sources is sometimes problematical as much of the first quarter of the article is taken up with what can appear to be an extended literature review. It is certainly necessary to demonstrate knowledge of the scholarly literature but it is also necessary to ‘digest’ this literature so that what comes across is the main thrust of the argument. This does not come across clearly enough. It is difficult to tell from the abstract or the introduction exactly how the author places her/himself within current scholarly debates and, therefore, what it is that the author is adding to scholarly knowledge/debate, what her/his argument is and how this goes beyond what other authors have said. This needs to be articulated much more clearly and forcefully and the literature review should be subordinate to it. Without it, the article, despite the care with which it has been researched, does not really do anything. What comes across is a great deal of knowledge that has been gathered and is held together by the theme of gender and devotion in the home but, importantly, is not sufficiently motivated in terms of argument. In some ways, it is possible that the author has attempted too much and the article may be more successful if it were to be streamlined.

Some things need to be explained more:

Caterina dei Francesi and Bonifacio Lupi. Some readers will immediately associate Lupi with Padua and so the link to Florence is not clear. Some reference to recent literature would be useful. I was unsure why no references were made to recent literature on the Meditationes – Holly Flora, Sarah McNamer.

Smaller issues:

Be aware of repetition. You don’t have to refer to Datini as ‘the merchant of Prato’ more than once. After that his name is sufficient. I, personally, find it slightly patronizing to refer to women only by their first name so I’d prefer Bartolomea degli Obizi rather than Bartolomea. (Should it be ‘Obizzi’ – this seems to be more common?) Some things which are titles (e.g. I frutti della ligna) are neither italicized nor are they put in inverted commas.

Author Response

Response to reviewer 2

Thank you for your review. I completely agree about the extended literature review. A previous reviewer insisted upon such a review, but of course, I need to situate myself within it. I hope that I have managed to do so now more effectively. A previous review also required that everything be cited on late female piety and domestic devotion , as my first attempt had taken for granted that the feminine nature of penitential and affective piety did not need to be explained, so I went too far the other way. I hope that I have sharpened my contrast of the two spaces, male and ecclesiastical, female and domestic, as a way of looking at domestic devotion as profoundly other to the clerical mentality.

I have added to my reference to Lupi. I have added Flora and McNamer to my references, but I did not want to discuss authorship of the Meditationes or dwell on that text in particular.

I have removed the ‘merchant of Prato’ from mentions of Datini. I agree completely about using first names for women. I was using the first name for Antoninus Pierozzi and also for Bernardino da Siena frequently, so I hope I was showing sloppiness rather than gender bias. I have changed Bartolomea to Bartolomea degli Obizzi, and in other instances after giving the full name have used surnames (e.g. Tornabuoni). I have left Antoninus and Bernardino alone, as that seems to be standard.

Round 2

Reviewer 3 Report

This is now clear and will be a good overview for those seeking to begin research into the use of the home as a spiritual space. The research is extensive.